# Enhanced electrocaloric efficiency via energy recovery

E. Defay [1,2,3,4], R. Faye[1], G. Despesse [2], H. Strozyk[1], D. Sette[1], S. Crossley [3,5], X. Moya [3] &
N.D. Mathur [3]

Materials that show large and reversible electrically driven thermal changes near phase transitions have been proposed for cooling applications, but energy efficiency has barely been explored. Here we reveal that most of the work done to drive representative electrocaloric cycles does not pump heat and may therefore be recovered. Initially, we recover 75–80% of the work done each time $BaTiO_3$-based multilayer capacitors drive electrocaloric effects in each other via an inductor (diodes prevent electrical resonance while heat flows after each charge transfer). For a prototype refrigerator with 24 such capacitors, recovering 65% of the work done to drive electrocaloric effects increases the coefficient of performance by a factor of 2.9. The coefficient of performance is subsequently increased by reducing the pumped heat and recovering more work. Our strategy mitigates the advantage held by magnetocaloric prototypes that exploit automatic energy recovery, and should be mandatory in future electrocaloric cooling devices.

[1] Materials Research and Technology Department, Luxembourg Institute of Science and Technology (LIST), 41 rue du Brill, Belvaux L-4422, Luxembourg. [2] CEA, LETI, Minatec Campus, Université Grenoble Alpes, 17 Rue des Martyrs, 38054 Grenoble, France. [3] Department of Materials Science, University of Cambridge, Cambridge CB3 0FS, UK. [4] Present address: Materials Research and Technology Department, Luxembourg Institute of Science and Technology (LIST), 41 rue du Brill, Belvaux L-4422, Luxembourg. [5] Present address: Department of Applied Physics, Stanford University, Stanford, CA 94305, USA. These authors contributed equally: E. Defay, R. Faye. Correspondence and requests for materials should be addressed to E.D. (email: emmanuel.defay@list.lu) or to N.D.M. (email: ndm12@cam.ac.uk)

It is important to pump heat in a wide range of scenarios, notably for cooling food, beverages, medicine, electronics, and habitable environments such as buildings and vehicles. Vapor-compression technology currently dominates the market, but involves harmful gases, is not suitable for miniaturization, cannot start up quickly, and can be noisy. Semiconducting Peltier technology does not suffer from any of these disadvantages, but is limited by low efficiency to niche applications[1]. An alternative option is to exploit reversible thermal changes that are typically driven near ferroic phase transitions by changes of magnetic field, electric field, or stress field[2]. The resulting magnetocaloric (MC), electrocaloric (EC), and mechanocaloric (mC) effects have now been exploited in over 40 MC prototypes[3], 10 EC prototypes[4–13], and 5 mC prototypes[14–18], where if present a regenerator such as a fluid column[19] permits the temperature difference between sink and load to greatly exceed the adiabatic temperature change $\Delta T$ that may be achieved in the caloric material alone.

It is common practice to reduce the work done to drive MC prototypes[3] by avoiding large changes of resultant flux density[20,21]. This scenario is achieved when a fixed magnetic field impinges upon part of a rotating annulus of MC material. Mechanical work is done (positive d$W$) to extract from the region of field an infinitesimal segment of the annulus that therefore cools. An irrecoverable component of this work is done to pump heat and overcome friction, but the remainder is automatically recovered (negative d$W$) by the simultaneous entry of another infinitesimal segment that therefore heats.

The above method of automatic energy recovery is not currently viable in EC prototypes because mechanically driven EC effects have not yet been demonstrated, despite reports of EC fluids[22,23]. Therefore, it is important to explore how to artificially recover as much as possible of the electrical work done (positive $W$) to charge and thus heat an EC capacitor when it subsequently discharges (negative $W$) and thus cools, cf. energy recovery for improving the efficiency of piezoelectric devices[24].

Here, we reveal that the electrical work done to drive large EC effects can be significantly greater than the cycle work required by the second law of thermodynamics to pump heat, inspiring the concept of energy recovery. We then demonstrate energy recovery using commercially available BaTiO$_3$-based multilayer capacitors (MLCs) to drive antiphase EC effects in each other via an inductor, reconciling the very different time scales on which charge and heat flow by using series diodes to prevent resonance. Last, we demonstrate a prototype EC refrigerator in which energy recovery enhances our coefficient of performance (COP) by a factor of 2.9, without reducing the temperature span of the device. We subsequently show that driving smaller EC effects increases our COP, at the cost of reducing the temperature span.

## Results

**Recoverable work**. For films that show giant EC effects[25–28], we will compare the driving work with the cycle work by assuming that the peak EC effect at optimum starting temperature $T_0$ can be reversibly driven in full at any nearby temperature (parallel isofield contours, Fig. 1a) (this is a plausible assumption, as giant EC effects arise with little hysteresis in relaxors, and near broad second-order transitions in films). The isothermal electrical polarization data used to deduce the giant EC effects of interest readily permits calculation[21,29] of the isothermal electrical driving work $|W|$ per unit volume of EC material at $T_0$, and leads us to consider Ericsson cooling cycles (1→2→3→4→1, Fig. 1b) in which isothermal EC effects are driven and undriven whenever the EC material lies alternately at each end of a thermal gradient in an ideal regenerator[19]. During operation, isothermal field application at the hot end causes the EC material to dump heat at $T_h$ to a

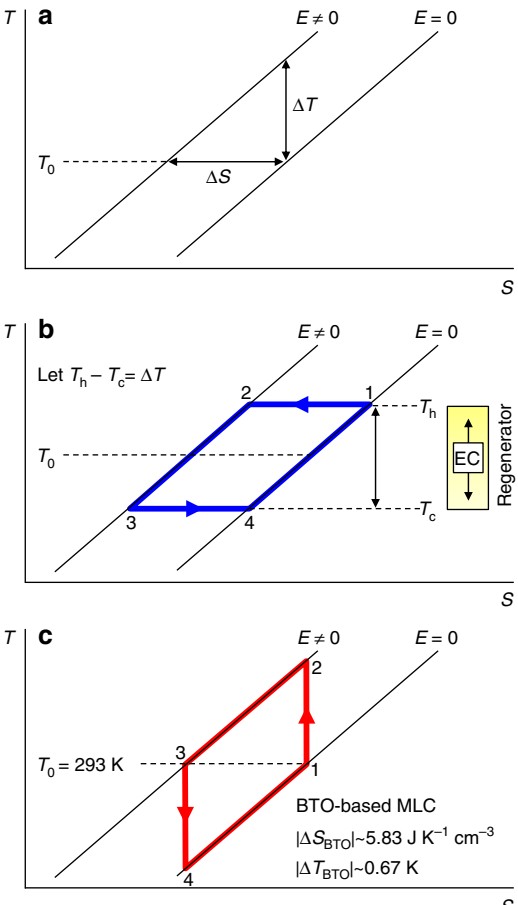

**Fig. 1** Simplified EC materials performance and resulting refrigeration cycles. **a** Near and at optimum starting temperature $T_0$, we assume that one can achieve the maximum adiabatic temperature change $\Delta T$, and the maximum isothermal entropy change $\Delta S$, due to a change of applied electric field $E$. For the performance schematised in **a**, we show **b** an Ericsson cycle (1→2→3→4) where a thermal gradient in a fluid regenerator permits the separation of $T_c$ (at which heat is absorbed) and $T_h$ (at which heat is expelled), and **c** the Brayton cycle (1→2→3→4) that we realize experimentally using two BTO-based MLCs, C1 and C2. Values for $|\Delta T_{BTO}|$ and $|\Delta S_{BTO}|$ describe the BTO layers alone

sink; isothermal field removal at the cold end causes the EC material to absorb heat at $T_c$ from a load; and the heat flowing from the EC material to the regenerator during the transit at finite field $\left(\int_2^3 T \mathrm{d}S\right)$ has the same magnitude as the heat flowing in the opposite direction during the transit at zero field $\left(\int_4^1 T \mathrm{d}S\right)$, implying true regeneration.

We will choose $T_h = T_0 + \Delta T/2$ and $T_c = T_0 - \Delta T/2$ purely because it is convenient to let the product of two material parameters $|\Delta S \Delta T|$ denote the Ericsson cycle work per unit volume of EC material ($\Delta S$ is isothermal entropy change, cycle work is identified from the cycle area in Fig. 1b, and we note that $|\Delta S \Delta T|$ does not represent the area of a Carnot cycle given that $\Delta S$ and $\Delta T$ both describe the full transition). Using the first and second laws of thermodynamics, cycle work $|\Delta S \Delta T|$ equals the work done to drive EC effects at $T_h$ minus the lesser work that could be recovered when undriving EC effects at $T_c$ (no electrical work is done at high field (2→3, Fig. 1b) as the polarization is constant under our assumption of peak performance (Fig. 1a)). For materials of interest, the cycle work $|\Delta S \Delta T|$ is much smaller

| EC material | $T_0$ (K) | $|\Delta E|$ (kV cm$^{-1}$) | $|\Delta S|$ (mJ K$^{-1}$ cm$^{-3}$) | $|\Delta T|$ (K) | $|\Delta S \Delta T|$ (J cm$^{-3}$) | $|W|$ (J cm$^{-3}$) | $|W|/|\Delta S \Delta T|$ | Ref. |
|---|---|---|---|---|---|---|---|---|
| PbZr$_{0.95}$Ti$_{0.05}$O$_3$ | 499 | 480 | 62.5 | 12 | 0.75 | 10.3 | 13.7 | 25 |
| 0.93PMN-0.07PT | 308 | 720 | 77.9 | 7.3 | 0.57 | 4.3 | 7.5 | 26 |
| P(VDF-TrFE) | 353 | 2000 | 107.6 | 12.5 | 1.35 | 5.2 | 3.8 | 27 |
| P(VDF-TrFE-CFE) | 310 | 3100 | 122.6 | 12.5 | 1.53 | 7.9 | 5.2 | 27 |
| *P(VDF-TrFE-CFE) | 350 | 3500 | 174.3 | 22 | 3.83 | 8.7 | 2.3 | 28 |

**Table 1 Recoverable work and cycle work for giant EC materials**

At starting temperature $T_0$, field change $|\Delta E|$ yields $|\Delta S|$ and $|\Delta T|$, and hence cycle work $|\Delta S \Delta T|$ for the Ericsson cycle of Fig. 1b. The isothermal electrical work[21,29] $|W|$ associated with $|\Delta S|$ at optimum starting temperature $T_0$ approximately represents the recoverable work at cold temperature $T_c$ (Fig. 1b) given that $|W| >> |\Delta S \Delta T|$. 0.93PMN-0.07PT = 0.93Pb(Mg$_{1/3}$Nb$_{2/3}$)O$_3$-0.07PbTiO$_3$. P(VDF-TrFE) = poly(vinylidene fluoride-trifluoroethylene) 55/45 mol%. P(VDF-TrFE-CFE) = poly(vinylidene fluoride-trifluoroethylene-chlorofluoroethylene) 59.2/33.6/7.2 mol%. *P(VDF-TrFE-CFE) = poly (vinylidene fluoride-trifluoroethylene-chlorofluoroethylene) 56.2/36.3/7.6 mol%.

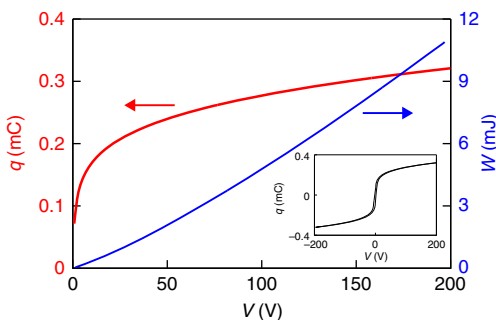

**Fig. 2** Electrical characterization of a poled MLC. On increasing applied voltage $V$ over 0.25 s to ensure adiabatic conditions (see Methods), charge $q$ was reversibly increased from its remanent value of ~0.1 mC by doing work $W$, such that an equivalent amount of electrical energy $E$ was stored in the capacitor. Inset: the corresponding major loop measured over 1 s

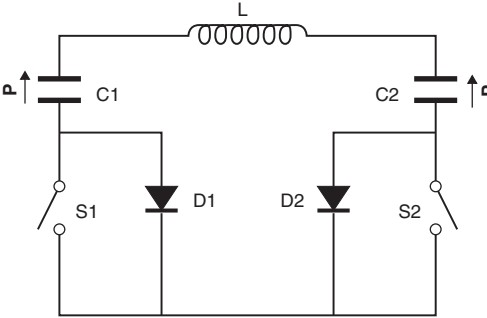

**Fig. 3** Circuit for the demonstration of energy recovery. Electrical energy was passed via inductor L between EC capacitors C1 and C2, which are poled ferroelectric MLCs with a variable polarization **P** of finite magnitude and fixed direction. Switches S1 and S2, and diodes D1 and D2, prevented resonance so that heat could flow between each charge-transfer event (see Methods). For simplicity, we do not show the Sourcemeter used to inject the initial charge into C1 and subsequently measure its voltage, or the Sourcemeter used to measure voltage across C2

than the isothermal work $|W|$ done to drive $\Delta S$ at $T_0$ (Table 1). Therefore, the magnitude of the work done at $T_h$, and the magnitude of the recoverable work at $T_c$, may both be approximately represented by $|W|$ evaluated at $T_0$. The prospect of recovering a relatively large amount of work $|W| >> |\Delta S \Delta T|$ motivates the following demonstration of energy recovery in two MLCs and then a prototype heat pump, but, more generally, the presence of any recoverable work in each and every cycle implies the need for energy recovery.

**EC multilayer capacitors**. Giant EC effects require large electric fields of ~100 V μm$^{-1}$, but breakdown fields of this magnitude have only been achieved in films no thicker than a few tens of microns, and such films alone cannot pump much heat. However, an assembly of films in the form of an MLC comprises a macroscopic quantity of sub-divided EC material[30], to which the interdigitated metallic electrodes provide good thermal and electrical access. MLCs based on giant EC materials are therefore under development[6,31], but they are not generally available, so we will exploit serendipitous EC effects in commercially available ferroelectric MLCs that are based on doped BaTiO$_3$ (BTO)[32].

Near room temperature, the poled MLCs that we use here show an adiabatic temperature change at the terminals of $|\Delta T_{MLC}| \sim 0.54$ K in response to a voltage change of $|\Delta V| = 200$ V (see later), implying that the active layers alone would show an adiabatic temperature change of $|\Delta T_{BTO}| \sim 0.67$ K and an isothermal entropy change of $|\Delta S_{BTO}| \sim 5.83$ mJ K$^{-1}$ cm$^{-3}$ (see Methods). For MLCs undergoing the Ericsson cycle of Fig. 1b near room temperature, this implies a cycle work of $|\Delta S_{BTO} \Delta T_{BTO}| \sim 3.93$ mJ cm$^{-3}$.

Under isothermal conditions at room temperature, the minimum (constant current) work done to charge one poled MLC from a remanent charge of $q(0 \text{ V}) \sim 0.1$ mC to $q(200 \text{ V}) \sim 0.3$ mC (red data, Fig. 2) is found via $dW = Vdq$ to be $W \sim 11.0$ mJ (blue data, Fig. 2), implying a stored energy of $E \sim 11.0$ mJ given that we observe good reversibility even under bipolar conditions (inset, Fig. 2). Normalizing by the active volume of EC material yields $W \sim 1.41$ J cm$^{-3}$, and hence a large value of $|W|/|\Delta S_{BTO} \Delta T_{BTO}| \sim 358$, such that $|W|$ approximately represents both the work done and the recoverable work, for MLCs that undergo the Ericsson cycle of Fig. 1b near room temperature. The discrepancy between recoverable work $|W|$ and cycle work $|\Delta S_{BTO} \Delta T_{BTO}|$ here is much larger than the corresponding discrepancy for giant EC materials (Table 1), and hence almost all of the work done to drive an MLC in the Fig. 1b Ericsson cycle could be recovered (the reader is reminded that energy recovery can be achieved in every single cycle, and is therefore worthwhile even if $|W|$ were to only just exceed $|\Delta S \Delta T|$).

**Energy recovery strategy**. The ideal destination for the electrical energy that is recovered from a discharging EC capacitor is a second EC capacitor that operates in antiphase, permitting quasi-continuous cooling[4,5]. After transferring charge between two such capacitors, a delay is required so that each capacitor can exchange heat. We will use an inductor of inductance $L = 2.2$ mH to mediate charge transfer between two poled ferroelectric MLCs (C1 and C2) that each possesses a non-linear capacitance of

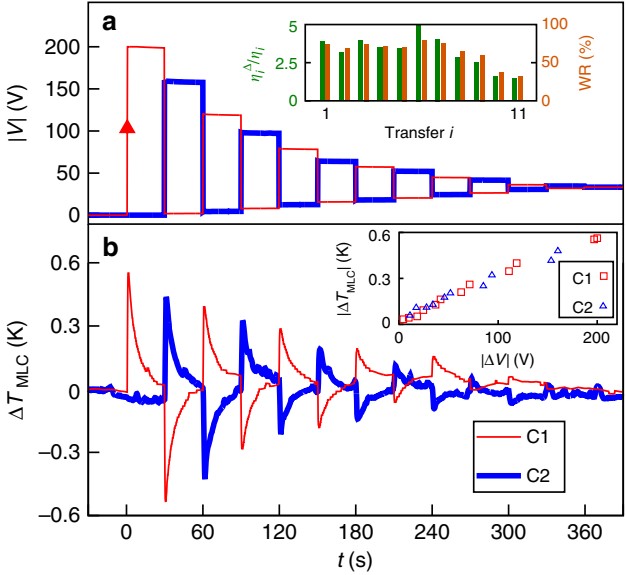

**Fig. 4** Demonstration of energy recovery. **a** Voltage magnitude $|V|$ across C1 (thin red line) and C2 (thick blue line) after charging C1 at time $t = 0$ (red arrow) and then transferring charge between C1 and C2 every 30 s (transfer $i$ took place at $t = 30i$ s). Inset: $\eta_i^\Delta/\eta_i = W_i/(W_i - W_{i+1})$ and WR $= 1 - \eta_i/\eta_i^\Delta = W_{i+1}/W_i$, where work $W_i$ denotes the work done in transfer $i$ by the MLC that discharges. **b** The corresponding changes of EC temperature $\Delta T_{\text{MLC}}$ for C1 (thin red line) and C2 (thick blue line). Inset: for C1 and C2, we plot the sharp jumps of $|\Delta T_{\text{MLC}}|$ in **b** vs. the corresponding jumps of $|\Delta V|$ in **a**

$C = dq/dV \leq 47\ \mu\text{F}$ (Fig. 2), and we will prevent electrical resonance by introducing switches (S1 and S2) and diodes (D1 and D2), such that charge transfer proceeds only on demand. Our circuit appears in Fig. 3, and the principle of operation is described in Methods. (While preparing this manuscript, we learned that inductors have been used to pass charge between capacitors in order to improve the efficiency of piezoelectric actuation[24], piezoelectric harvesting[33], and pyroelectric harvesting[34].)

As explained in Methods, each charge transfer between the MLCs proceeds so quickly that the resulting EC effects are adiabatic, whereas isothermal EC effects would require an impractically large inductance. Therefore, we will demonstrate energy recovery using two MLCs that undergo Brayton cycles (1→2→3→4→1, Fig. 1c) rather than the Ericsson cycles of Fig. 1b. Most of the driving work in our Brayton cycles can be recovered because $|W|/|\Delta S_{\text{BTO}} \Delta T_{\text{BTO}}| \sim 358$ is large. This ratio is unchanged with respect to the Ericsson cycle because the cycle area remains $|\Delta S_{\text{BTO}} \Delta T_{\text{BTO}}|$, and because $|W|$ is very similar under both isothermal and adiabatic conditions (see Methods).

**Demonstration of energy recovery.** After initially charging C1 to 200 V at time $t = 0$ and letting the resulting EC heat leak out over the next ~30 s, the two MLCs exchanged charge quickly every ~30 s (Fig. 4a), and thus underwent antiphase Brayton cycles (Fig. 4b). Each charge transfer produced an adiabatic decrease of temperature $\Delta T_{\text{MLC}} < 0$ in the MLC that lost charge, and an adiabatic increase of temperature $\Delta T_{\text{MLC}} > 0$ in the MLC that gained charge (Fig. 4b), as measured at the MLC terminals that exchanged heat with the active EC layers on a fast time scale ($\tau_{\text{th}}^{\text{int}} \sim 0.2$ s, Supplementary Note 1). Following these fast changes of temperature, each MLC slowly returned to ambient over ~30 s due to the exchange of heat with the laboratory bench via the thin electrical wiring ($\tau_{\text{th}}^{\text{ext}} \sim 8$ s, Supplementary Note 1).

Ideally, all of the charge that was initially injected into C1 would continue to be transferred between the two MLCs, but the jump in voltage $\Delta|V| > 0$ when one MLC gained charge was not as large as the jump in voltage $\Delta|V| < 0$ when the other MLC lost charge, reflecting incomplete transfer between our non-linear capacitors due to Joule losses in the inductor and the active diode (charge conservation is confirmed via $q(V)$ (Fig. 2)). The high-voltage and low-voltage plateaux therefore converged as a result of each charge transfer, but they also showed a small convergence between transfers due to leakage through the active diode (and not the MLCs, for which a leakage current of <1 nA at 200 V would imply a temperature increase of <5 mK over 30 s).

As the charge-transfer process proceeded, the magnitude of the voltage jump $|\Delta V|$ and the resulting temperature change $|\Delta T_{\text{MLC}}|$ both decreased in an approximately linear manner (inset, Fig. 4b). After all 11 transfers, the final voltage $|V| \sim 33$ V across each MLC corresponded to a final charge of $q(33\text{ V}) \sim 0.2$ mC (Fig. 2), confirming that the injected charge of $q(200\text{ V}) - q(0\text{ V}) \sim 0.2$ mC had been ultimately shared between each MLC, where it augmented the remanent charge of $q(0\text{ V}) \sim 0.1$ mC.

**Analysis of demonstration.** We will quantify the effectiveness of our energy recovery process in terms of the enhancement it confers on materials efficiency[21,29] $\eta = |Q/W|$, which compares the heat $Q$ and the work $W$ for reversible EC effects that are driven once under isothermal conditions (this differs from the COP for a material, which is defined in terms of the net work done to complete a cooling cycle[35]). The isothermal heat $Q_{\text{BTO}} > 0$ absorbed per unit volume of EC material will be deduced (see Methods) from our measurements of the adiabatic temperature change $\Delta T_{\text{MLC}} < 0$ that results from the discharge in transfer $i$ (Fig. 4), and this heat will be presented as $Q_i$ rather than as $Q_{\text{BTO}}$. The isothermal work done to produce these EC effects will be deduced from the adiabatic changes of driving voltage (Fig. 4a) using the adiabatic relation $W(V)$ (Fig. 2), which is reasonable given the similarity of $|W|$ under both isothermal and adiabatic conditions (see Methods).

We will compare $|Q_i|$ for the MLC that cools in transfer $i$ with the net energy that is in effect consumed by this transfer, that is, the work $W_i$ done to achieve transfer $i$ less the smaller work $W_{i+1}$ that is available later for transfer $i+1$. The one-shot efficiency $\eta_i = |Q_i/W_i|$ for the cooling event of transfer $i$ is thus enhanced to yield $\eta_i^\Delta = |Q_i/(W_i - W_{i+1})|$, such that we have $\eta_i^\Delta/\eta_i = W_i/(W_i - W_{i+1})$. For the first several transfers, we find $\eta_i^\Delta/\eta_i \sim 4$ (inset, Fig. 4a), implying that the work recovered was WR $= W_{i+1}/W_i = 1 - \eta_i/\eta_i^\Delta \sim 75\%$ (inset, Fig. 4a). In each of these early transfers, 75% of the work done and subsequently stored as energy was therefore then used to do work in the next transfer (work itself cannot be stored). One may equally parameterize the energy stored between transfers by defining the energy recovered ER = WR, but here we focus instead on the driving work.

After a sufficient number of transfers, the two MLCs exchanged less charge. Consequently, the peak transfer current (Supplementary Note 2) fell below the 3 A rating of our diodes, whose 0.7 V threshold for operation was eventually no longer reached. We therefore observed maxima in $\eta_i^\Delta/\eta_i$ and WR for $i = 6$, both here (inset, Fig. 4a) and in every subsequent trial that we performed.

After all 11 transfers, the ~33 V across each MLC implied a total stored energy of $E \sim 2.4$ mJ in the two MLCs, as determined via $W(V)$ (Fig. 2). This represents ~22% of the original $E \sim 11.0$ mJ when C1 was charged to 200 V, such that almost one-quarter of the input energy remained unspent, and could in principle be recovered. Most of the lost input energy was

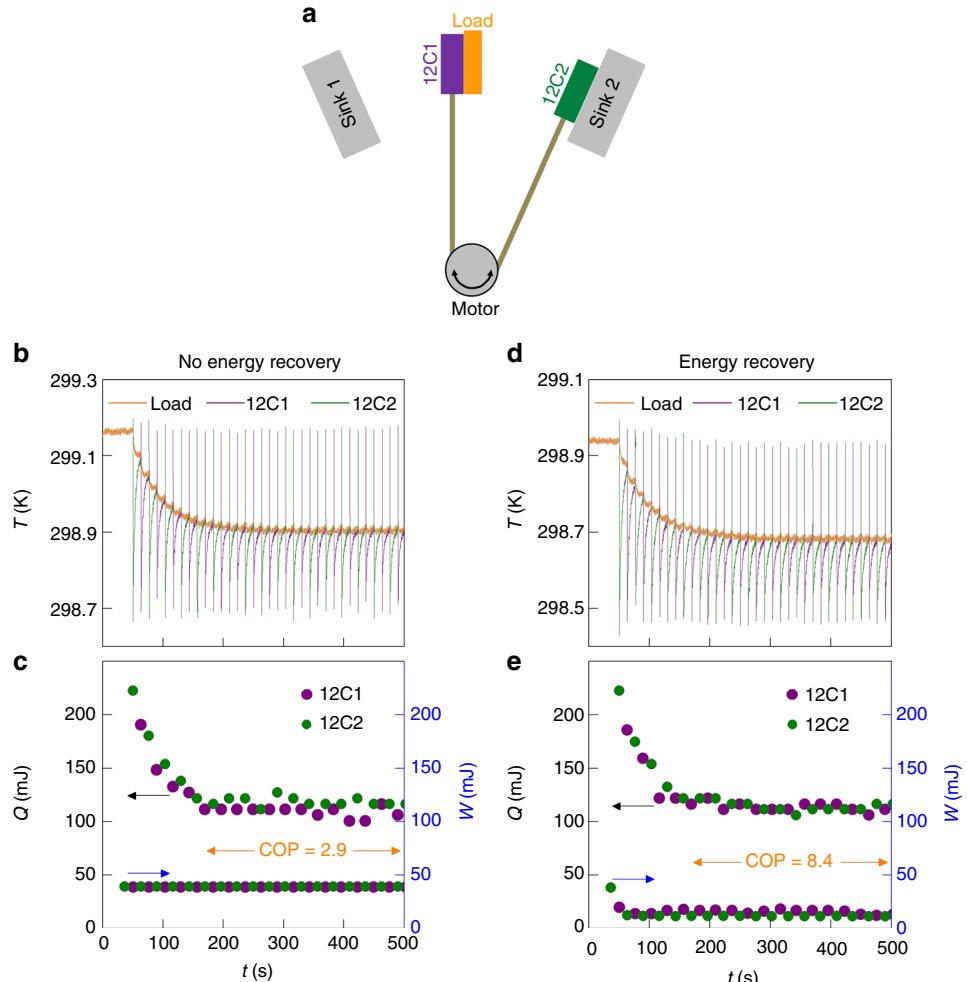

**Fig. 5** EC prototype without and with energy recovery. **a** Device schematic with EC plates 12C1 and 12C2, each based on 12 MLCs. Data are presented (**b**, **c**) without energy recovery and (**d**, **e**) with energy recovery. **b**, **d** In a given half cycle, each plate (whose temperature is given by the green or purple data) underwent rapid adiabatic EC cooling before slowly absorbing heat from the load, whose temperature $T_c$ (orange data) fell by 0.26 K with respect to starting temperature $T_h$ on reaching the steady state around 250 s after start-up at time $t = 40$ s. **c**, **e** For each half cycle, we show the heat $Q$ absorbed by the cold plate from the load, and the work $W$ done to charge the other plate back to 70 V at the sink (where it thus dumps heat)

dissipated by the Joule heating in the inductor and the active diodes during the transfers, such that only ~80% of the work $W_i$ done by the MLC that lost charge during transfer $i$ was added as energy $E_i$ to the MLC that gained charge, even for our largest values of $i$. However, ~4% of the input energy was lost during all of the wait times between transfers, via the leakage through the active diode rather than internal MLC discharge, as explained earlier. These losses between transfer events, which are barely perceptible as gradients in the high-voltage and low-voltage plateaux of Fig. 4a, start at 2% for the first plateau, and fall to 0.1% for the last plateau (the initial figure of 2% implies that 98% of the energy $E_1$ added to C2 during transfer 1 was available to do work $W_2$ on C1 during transfer 2).

An alternative means of parameterizing the energy recovery demonstrated here is to evaluate the total heat $\Sigma_{i=1}^{i=11}|Q_i|$ associated with the 11 identifiable cooling events in the two MLCs, all of which were in effect driven by doing work $W_1$ at time $t = 0$. This enhances the one-shot efficiency $\eta_i = |Q_i/W_i|$ for the cooling event associated with transfer $i = 1$ in order to yield $\eta^\Sigma = \sum_{i=1}^{i=11}|Q_i/W_1|$, such that $\eta^\Sigma/\eta_1 = \Sigma_{i=1}^{i=11}|Q_i/Q_1| \sim 5$. This implies that WR $= 1 - \eta_1/\eta^\Sigma \sim 80\%$ (Supplementary Note 3), which is similar to WR $= 1 - \eta_i/\eta_1^\Delta \sim 75\%$ from above, as

expected given the approximate equivalence between $\eta^\Sigma/\eta_1 \sim 5$ and $\eta_i^\Delta/\eta_i \sim 4$ (Supplementary Note 4).

**Prototype with energy recovery**. We will now demonstrate artificial energy recovery in a fully automated prototype cooling device, where two EC plates that each comprises 12 MLCs were operated in antiphase (schematic in Fig. 5a, photograph in Supplementary Note 5, video in Supplementary Movie 1). The construction, circuit (diagram in Supplementary Note 6), timing, and monitoring are described in Methods. As before, charge transfer took place via an inductor, with series diodes to prevent resonance. The maximum voltage change across a plate was limited to 70 V in order to avoid any possibility of breakdown. The laboratory air conditioning was switched off in order to minimize thermal drift.

For many cycles of steady-state operation, where heat $Q$ is pumped from the load in each half cycle by doing work $W$ in the previous half cycle, we will identify the COP for our prototype as the total heat pumped from the load divided by the total work done, such that COP $= \overline{Q}/\overline{W}$. (This is discussed further in Methods, along with the fact that COPs calculated from values of

heat and work represent upper bounds on device COPs.) Initially, we will identify the steady-state COP without energy recovery. We will then improve it by exploiting energy recovery in order to reduce the net work done, without reducing the net heat pumped in order to preserve device temperature span. Last, we will demonstrate improvements to our steady-state COP by reducing both the net work done and the net heat pumped. The work associated with a given change of voltage will be evaluated using electrical data obtained for each plate, but it could have been calculated with reasonable precision using the electrical data for a single MLC via $12W(V)$ (Fig. 2).

Operation without energy recovery was as follows. The two plates underwent antiphase Brayton cycles ($T_h \neq T_c$), during which they were translated using a micromotor. Consequently, each plate transferred heat from a common load at cold temperature $T_c$ to a dedicated sink at a common hot temperature $T_h$. In practice, the value of $T_h$ was nominally unchanged with respect to the starting temperature because each sink was sufficiently massive. In one full cycle, a given plate in thermal equilibrium with its sink was translated to the load, underwent adiabatic EC cooling via wasteful discharge ($70\,V \rightarrow 0\,V$), absorbed heat $Q$ from the load to reach thermal equilibrium, was translated to its sink, underwent adiabatic EC heating by doing work $W$ ($0\,V \rightarrow 70\,V$), dumped heat to its sink to reach thermal equilibrium, and was translated back to the load to complete the cycle.

The resulting performance without energy recovery was as follows. Figure 5b shows the time evolution of the load temperature (orange data) after start-up, as well as the corresponding temperatures of 12C1 (purple data) and 12C2 (green data) at the times when they were in contact with the load and thus in the infrared (IR) camera field of view. After turn on, each nominally identical EC cooling step of $|\Delta T| \sim 0.47\,K$ in 12C1 and 12C2 was followed by an increase of plate temperature (green and purple data between sharp drops) due to the absorption of heat $Q$ from the load (Fig. 5c). In the steady state at around 250 s after start-up, an average value of $\overline{Q} \sim 113 \pm 6\,mJ$ was identified by multiplying the average $\sim 0.21 \pm 0.01\,K$ increase of plate

temperature with plate heat capacity $\gamma = 0.53\,J\,K^{-1}$ (see Methods), and the load temperature $T_c$ lay $T_h - T_c \sim 0.26\,K$ below starting temperature $T_h$.

In each half cycle without energy recovery, the work $W(0 \rightarrow 70\,V) = I \int_{t(0\,V)}^{t(70\,V)} \overline{V}(t')dt'$ done to charge a given plate was identified by averaging ten measurements of voltage $V$ against time $t$ at the charging current of $I = 10\,mA$ (Supplementary Note 7). The work done to fully charge 12C1 and 12C2 to 70 V yielded an average value of $\overline{W} \sim 39 \pm 0.2\,mJ$ (Fig. 5c). Given also the average value of $\overline{Q} \sim 113 \pm 6\,mJ$ that we report above for each half cycle, we identify a COP of $\overline{Q}/\overline{W} \sim 2.9 \pm 0.13$ (Fig. 5c) for our prototype with $T_h - T_c \sim 0.26\,K$ in the steady-state without energy recovery.

To implement energy recovery in our prototype, the aforementioned EC effects in each half cycle were primarily driven by transferring charge from one plate to the other. The transfer of charge was then in effect completed by discharging the plate whose voltage was low after transfer ($V_{lat} \rightarrow 0\,V$), and charging the plate whose voltage was high after transfer ($V_{hat} \rightarrow 70\,V$). Supplementary Note 8 shows the time dependence of the voltage across each plate during operation. Both of the two-step changes of voltage ($70\,V \rightarrow V_{lat} \rightarrow 0$ and $0 \rightarrow V_{hat} \rightarrow 70\,V$) were completed over a relatively short time (1 s) with respect to the 13 s heat-flow interval during which they fell (the reader is reminded that timing is described in Methods).

The value of $V_{lat} = 1.1\,V$ was so small that the subsequent plate discharge produced no further EC cooling of note, and therefore our in operando measurements of load and plate temperatures (Fig. 5d) were nominally unmodified with respect to their counterparts without energy recovery (Fig. 5b), such that with energy recovery we obtained similar average values of $\overline{Q} \sim 114 \pm 4\,mJ$ (Fig. 5e) and $T_h - T_c \sim 0.26\,K$. By contrast, finite values of $V_{hat} \sim 50\,V$ (identified from $V(t)$ in Supplementary Note 8) permitted a reduced work $W(V_{hat} \rightarrow 70\,V) = I \int_{t(V_{hat})}^{t(70\,V)} \overline{V}(t')dt'$ (evaluated via $V(t)$ and $W(V)$ in Supplementary Note 7) of average value $\overline{W} \sim 14 \pm 2.5\,mJ$ (Fig. 5e) to fully charge the partially charged plate to 70 V (and dump further EC heat to the active sink). By thus recovering $ER \sim (39 - 14)/39 \sim 65\%$ of the

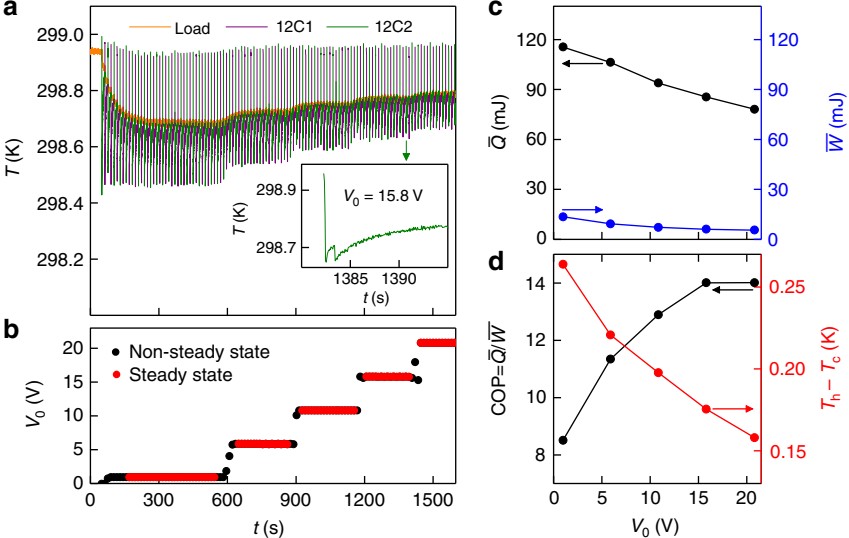

**Fig. 6** EC prototype with modified energy recovery. **a** Data shown in the first ~600 s are similar to the data of Fig. 5d. On subsequently setting (**b**) finite values of $V_0$, each plate (whose temperature is given by the green or purple data) underwent adiabatic EC cooling in two separate steps, after each of which it slowly absorbed heat from the load, whose temperature $T_c$ (orange data) was thus reduced with respect to starting temperature $T_h$. The inset shows a detail of the two-step process for 12C2. For steady-state operation at different values of $V_0$ (red in **b**), we present (**c**) average values of heat $\overline{Q}$ and work $\overline{W}$, and hence (**d**) values of COP = $\overline{Q}/\overline{W}$ for each temperature span $T_h - T_c$ identified via **a**. Note that $V_0 = 0$ prior to initial steady-state operation with $V_0 = 0.9\,V$

energy stored by having done work to fully charge a given plate to 70 V (Supplementary Note 7), the COP of our prototype was improved by a factor of $(1 - \mathrm{ER})^{-1} \sim 2.9$ to yield a value of $\overline{Q}/\overline{W}$ ~ 8.4 ± 1.0 (Fig. 5e) for the same value of $T_{\mathrm{h}} - T_{\mathrm{c}} \sim 0.26$ K.

We will now modify the energy recovery process as follows. Instead of driving the EC plates of the prototype between 0 and 70 V, we will employ finite start and finish voltages of $V_0$ and 70 V for our two-step processes. Although finite start and finish voltages reduce $\overline{Q}$ by reducing the magnitude of the EC effects in each plate, we will see that they reduce $\overline{W}$ even further. (We note that elsewhere in the EC literature[25], finite start and finish voltages have been exploited to avoid the low-field antiferro-electric regime of $\mathrm{PbZr_{0.95}Ti_{0.05}O_3}$.)

When operating (Fig. 6a) at finite values of $V_0$ (Fig. 6b), the two-step discharge of a given plate (70 V → $V_{\mathrm{lat}}$ → $V_0$) produced two distinct adiabatic cooling steps (that are only seen separately on an expanded time scale, inset of Fig. 6a), while the concomitant two-step charging ($V_0$ → $V_{\mathrm{hat}}$ → 70 V) was inferred to produce two distinct heating steps away from the IR camera field of view. On increasing $V_0$, there was a reduction in the sum of the two adiabatic cooling steps for a given plate (as the reduction in the first cooling step exceeded the increase in the second cooling step, Supplementary Note 9), such that there was a reduction in the average heat $\overline{Q}(V_0)$ (Fig. 6c) for the device during steady-state operation (red data, Fig. 6b).

The concomitant reduction in $\overline{W}(V_0)$ (Fig. 6c) was identified during steady-state operation by calculating the top-up work $W(V_{\mathrm{hat}} \to 70\,\mathrm{V}) = I\int_{t(V_{\mathrm{hat}})}^{t(70\,\mathrm{V})} \overline{V}(t')\mathrm{d}t'$ for each plate as before (via $V(t)$ and $W(V)$ in Supplementary Note 7), where $V_{\mathrm{hat}}$ was obtained from $V(t)$ as measured for each plate during operation (Supplementary Note 10). Values of $\mathrm{COP}(V_0) = \overline{Q}(V_0)/\overline{W}(V_0)$ increase on increasing $V_0$ (Fig. 6d), and tend to saturate at $V_0 = 15.8$ V to a value of 14 ± 3.2 (Supplementary Note 11 shows $\overline{Q}(V_0)$, $\overline{W}(V_0)$ and $\mathrm{COP}(V_0)$ for each plate separately). Finite values of $V_0$ therefore increase the COP, at the cost of reducing $\overline{Q}$ (Fig. 6c) and thus the temperature span $T_{\mathrm{h}} - T_{\mathrm{c}}$ (Fig. 6d).

## Discussion

The work presented in this paper involved three major steps. First, we identified that it is possible to recover a significant amount of the electrical work done when driving EC cooling cycles. Second, we demonstrated how to recover 75–80% of the recoverable work done when transferring charge between two MLCs, and we avoided resonance so that heat could flow after each charge transfer. Third, we exploited our strategy for energy recovery in a prototype cooling device. We initially used our prototype to recover 65% of the work done without reducing the average heat $\overline{Q}$ pumped, such that our COP was increased by a factor of 2.9 without reducing temperature span $T_{\mathrm{h}} - T_{\mathrm{c}}$. Recovering more work (86%) led to an increase of COP at the cost of reducing $\overline{Q}$ and thus $T_{\mathrm{h}} - T_{\mathrm{c}}$.

In terms of materials efficiency[21,29], recovering 75–80% of the work done renders electrically driven EC effects just as good as mechanically driven MC effects that benefit from the pre-existing field of permanent magnets[20,21]. In terms of applications, artificial energy recovery in EC prototypes mitigates the advantage held by MC prototypes, which can automatically recover all recoverable work, as explained earlier. In future, artificial energy recovery in EC prototypes could prove particularly important given that MC materials and permanent magnets remain expensive, while EC ceramics and polymers tend to be cheap to both fabricate and drive.

## Methods

**Multilayer capacitors**. Our MLCs (MULTICOMP MC1210F476Z6R3CT) possessed 137 EC layers (1.95 mm × 3.2 mm × 9.1±1 μm) between 138 interdigitated Ni electrodes of thickness ~1.5 μm. EDX confirmed that the EC layers were based on $\mathrm{BaTiO_3}$ doped with detectable levels of Zr and Ca. All MLCs used in this study were ferroelectric and electrically poled.

**MLC mounting for the demonstration of energy recovery**. The two MLCs were suspended above the laboratory bench by their thin electrical wiring, and shielded from air currents using an overturned plastic box. EC temperature change in each MLC was measured using an electrically isolated Pt-100 thermometer attached to one terminal using Apiezon N grease.

**Charging time and work done to drive EC effects in MLCs**. Using a Keithley High Voltage Sourcemeter 2410 configured as a current source, we established that the work $|W|$ done to charge a poled MLC to 200 V varied by just ±2% when varying the charging current from 100 nA to 1 mA in order to vary the charging time from 2200 s to 0.24 s. Charging over 2200 s was isothermal given that an infrared camera (X6580sc, FLIR) detected no temperature change to within a resolution of 5 mK. Charging times of <1 s were adiabatic given that they yielded a single value of $|\Delta T_{\mathrm{MLC}}|$, as measured using the same camera. (Note that camera output was consistent with measurements made using the Pt-100 thermometer.)

**Electrical characterization of MLCs**. A Radiant Precision Premier II Ferroelectric Tester was used to apply a triangular voltage pulse, such that changes of 200 V took 0.25 s, and were therefore adiabatic. The maximum voltage of magnitude 200 V corresponds to ~222 kV cm$^{-1}$. The data are presented as $q(V)$ (Fig. 2 and inset) rather than displacement vs. field, in order to avoid errors associated with measuring the internal geometry.

**Parameterization of EC effects in MLCs**. The measured value of $|\Delta T_{\mathrm{MLC}}|$ ~ 0.54 K (Fig. 4b) implies $|\Delta T_{\mathrm{BTO}}| = 0.67$ K, if we assume internal thermalization[32] such that $\Delta T_{\mathrm{MLC}} = \beta \Delta T_{\mathrm{BTO}}$. Here, $\beta = c_{\mathrm{BTO}}\rho_{\mathrm{BTO}}V_{\mathrm{BTO}}/(c_{\mathrm{BTO}}\rho_{\mathrm{BTO}}V_{\mathrm{BTO}} + c_{\mathrm{Ni}}\rho_{\mathrm{Ni}}V_{\mathrm{Ni}}) = 0.80$ was evaluated for EC and electrode layers of pure BTO and pure Ni, with total volumes of $V_{\mathrm{BTO}} \sim 7.8$ mm$^3$ and $V_{\mathrm{Ni}} \sim 1.3$ mm$^3$, specific heat capacities of $c_{\mathrm{BTO}} \sim 434$ J K$^{-1}$ kg$^{-1}$ and $c_{\mathrm{Ni}} \sim 429$ J K$^{-1}$ kg$^{-1}$, and densities of $\rho_{\mathrm{BTO}} \sim 5840$ kg m$^{-3}$ and $\rho_{\mathrm{Ni}} \sim 8907$ kg m$^{-3}$ (the small amount of additional material around the active region is ignored). Given that MLCs show a good equivalence between directly and indirectly measured EC effects[32], we may write $|Q_{\mathrm{BTO}}| \sim -c_{\mathrm{BTO}}\rho_{\mathrm{BTO}}|\Delta T_{\mathrm{BTO}}| = 1.71$ J cm$^{-3}$ and $|\Delta S_{\mathrm{BTO}}| = |Q_{\mathrm{BTO}}|/T_0 \sim 5.83$ mJ K$^{-1}$ cm$^{-3}$ for $T_0 = 293$ K.

**Electrical circuit for demonstration of energy recovery**. During operation, the poled MLCs (C1 and C2) were both charged in the same sense as their remanent ferroelectric polarizations. Inductor L (1422509C, Murata) of inductance $L = 2.2$ mH and series resistance ~1 Ω was air filled to avoid the losses that would arise in a magnetic core. Diodes D1 and D2 (1N5406, Multicomp) were rated at 3 A. A Keithley High Voltage Sourcemeter 2410 configured as a voltage source was used to charge C1 to 200 V at time $t = 0$. To measure voltage across C1, this Sourcemeter was then reconfigured as a voltmeter with a high input impedance of nominally 10 GΩ. The use of an equivalent Sourcemeter permitted voltage to be measured across C2. Switches S1 and S2 were operated manually every ~30 s.

**Time constants for demonstration of energy recovery**. The rapid charge transfer between C1 and C2 was governed by electrical time constant $\tau_{\mathrm{el}} << \pi(LC/2)^{1/2} = 723$ μs, where $C = \mathrm{d}q/\mathrm{d}V|_{V=0\,\mathrm{V}} = 47$ μF represents an over-estimate due to the non-linearity in $q(V)$ (Fig. 2), and we ignore the effect of losses that render the charge transfer incomplete.

The thermal time constant $\tau_{\mathrm{th}} \sim \tau_{\mathrm{th}}^{\mathrm{int}} + \tau_{\mathrm{th}}^{\mathrm{ext}}$ for heat flow between the EC layers and some external point depends on both the internal time constant $\tau_{\mathrm{th}}^{\mathrm{int}}$ for heat flow inside the MLC (between EC layers and MLC terminals), and the external time constant $\tau_{\mathrm{th}}^{\mathrm{ext}}$ for heat flow outside the MLC (between MLC terminals and the external point of interest).

Even if we greatly overestimate $\tau_{\mathrm{el}}$ using our highest value of $C = \mathrm{d}q/\mathrm{d}V|_{V=0\,\mathrm{V}} = 47$ μF (Fig. 2), and even if we greatly underestimate $\tau_{\mathrm{th}}$ by assuming $\tau_{\mathrm{th}} = \tau_{\mathrm{th}}^{\mathrm{int}} \sim 0.2$ s (Supplementary Note 1), then an unfeasibly large value of $L \sim 172$ H would be required just to achieve $\tau_{\mathrm{el}} = \tau_{\mathrm{th}}^{\mathrm{int}}$. In practice, realistic values of $L << 1$ H force $\tau_{\mathrm{el}} << \tau_{\mathrm{th}}$, such that the EC effects driven here are highly adiabatic.

**Operating principle for demonstration of energy recovery**. With S1 and S2 open at time $t = 0$, a single shot of electrical energy was fed into the two-capacitor system by charging C1 to 200 V in the same sense as its remanent polarization (Figs. 2 and 4). Most of the EC heat generated in C1 subsequently flowed to the environment over ~30 s, such that the starting temperature was approximately recovered at $t = 30$ s.

To initiate transfer 1 at $t = 30$ s, S1 was closed such that a current flowing clockwise via D2 (electrons flowed anticlockwise) tended to discharge C1, and charge C2 in the same sense as its remanent polarization. This first transfer of charge resulted in C1 cooling and C2 heating on time scale $\tau_{\mathrm{th}}^{\mathrm{int}} \sim 0.2$ s (Supplementary Note 1). As soon as these EC effects had been initiated, heat also began to flow (primarily via the wiring) between each MLC and the environment

over ~30 s. The starting temperature was therefore approximately recovered at $t = 60$ s.

To initiate transfer 2 at $t = 60$ s, S1 was opened and then S2 was closed. This led to charge transfer from C2 to C1 via D1, resulting in each MLC showing thermal changes of opposite sign to before. Subsequent charge transfer every 30 s was achieved by opening the closed switch and then closing the open switch, causing each MLC to become alternately hot and cold. For each successive transfer, charge transfer and temperature change were reduced in magnitude until an equal share of the charge injected at $t = 0$ augmented the remanent charge on each MLC.

**MLC plates in EC prototype.** The two EC plates (12C1 and 12C2) each comprised 12 poled MLCs, whose terminals were soldered together via a fine Cu mesh on each side. The heat capacity $\gamma = 0.53 \, \text{J K}^{-1}$ of each plate was calculated by summing the heat capacities of (1) the 12 MLCs with total heat capacity $12(c_{\text{BTO}}\rho_{\text{BTO}}V_{\text{BTO}} + c_{\text{Ni}}\rho_{\text{Ni}}V_{\text{Ni}}) = 0.45 \, \text{J K}^{-1}$, (2) the 0.3 g of Sn60/Pb40 solder with specific heat capacity $177.5 \, \text{J K}^{-1} \, \text{kg}^{-1}$, and (3) the 0.07 g of Cu with specific heat capacity $385 \, \text{J K}^{-1} \, \text{kg}^{-1}$. Each plate was connected by a wooden stick to a motor (Standard servo, Parallax) that translated the plates between two copper heat sinks and one copper heat load. Repeatable thermal contact was facilitated by a metal oxide particle-containing silicone paste (RS 554-311, Radiospares) of thermal conductivity $0.65 \, \text{W m}^{-1} \, \text{K}^{-1}$.

**Electrical circuit for EC prototype.** This circuit (Supplementary Note 6) differs from the circuit used to demonstrate energy recovery (Fig. 3) in three key ways. First, we used a Keithley 2410 Sourcemeter to supply 10 mA in order to fully charge (no energy recovery) or top-up (with energy recovery) the plate to be fully charged to 70 V in a given half cycle, and we used a second such Sourcemeter to fully discharge (no energy recovery) or partially discharge (with energy recovery) the other plate (these intended voltage limits of 0 and 70 V were in practice set by the Keithley sourcemeter to values of 0.9 and 69.6 V, in order to avoid any overshoot). Second, we used an Arduino micro-controller to synchronize motor activity with the operation of relay switches (HFD2/005-M, RS Pro). Third, we used a larger 4.4 mH inductor that was formed from a coil of copper wire with series resistance 0.7 Ω.

**EC prototype timing without energy recovery.** The cycle period was 26.46 s. Each 13.23 s half cycle comprised three consecutive steps. (1) During the initial interval of 0.12 s, the two plates were simultaneously translated, such that the plate in thermal equilibrium with the load was translated to make contact with its sink, and the plate in thermal equilibrium with its sink was translated to make contact with the load. (2) A wait time of 0.11 s then elapsed. (3) Two processes took place simultaneously during the remaining 13 s: (i) the hot plate now in contact with the load underwent rapid adiabatic EC cooling over 0.1 s before slowly absorbing heat from load to reach thermal equilibrium, and (ii) the cold plate now in contact with the sink underwent rapid adiabatic EC heating (by applying 10 mA over 0.3 s) before slowly dumping heat to its sink to reach thermal equilibrium. (We also attempted to trigger EC effects in the plates prior to contact with the sink and load, which resulted in similar performance, but monitoring and reproducibility were compromised so this strategy was not pursued any further.)

**EC prototype timing with energy recovery.** Steps (1) and (2) above were preserved, while Step (3) above was revised to yield the following three-step process (a–c) without changing its 13 s duration (in what follows, either $V_0 = 0$ (Fig. 5d, e) or $V_0 \neq 0$ (Fig. 6), and the small charge-transfer times that we give for $V_0 = 0$ are reduced when $V_0$ is finite). (a) During an initial period of ≤0.2 s, there was a rapid and incomplete transfer of charge between the plate at 70 V and the plate at $V_0$, resulting in adiabatic EC effects of opposite sign and thus the onset of heat flow (from the plate thus charged to its sink, and to the plate thus cooled from the load). (b) After 0.5 s had elapsed with respect to the start of (a), heat flow from the plate that was partially charged in Step (a) was enhanced by further adiabatic EC heating (as a consequence of topping-up the charge to reach 70 V at 10 mA over ≤0.2 s), after which the plate continued to dump heat to its sink until reaching thermal equilibrium. (c) After 1 s had elapsed with respect to the start of (a), heat flow from the plate that was partially discharged in Step (a) was enhanced by further adiabatic EC cooling (as a consequence of being discharged via the dedicated resistor to reach $V_0$ over ≤0.1 s), after which the plate continued to absorb heat from the load until reaching thermal equilibrium.

**EC prototype monitoring.** The IR camera was used to measure the temperature of the load and the MLC plate nearby, sampling a rectangular region that occupied most of the top surface of each component. This top surface was painted black to increase emissivity to near unity. Precise emissivity values were identified by assuming a homogenous starting temperature.

**Prototype COP.** Given that work $W$ is done in one half cycle (EC heating) to pump heat $Q$ in the next half cycle (EC cooling), there would be an ambiguity if one were to calculate COP = $Q/W$. This is because one could choose to divide values of $Q$ and $W$ that describe either the same half cycle and different plates, or different half cycles and the same plate.

The COP that we evaluate from values of heat and work is not a true device COP for two reasons. First, heat is only pumped from our load towards our sink in order to compensate for heat leaking into our load from the environment, but with appropriate thermal isolation one could instead pump heat from a target item rather than the environment. Second, the work does not include what is done to drive the motor, the power supplies and the micro-controller, because we have not minimized the avoidable components of this work. This type of omission is standard practice with MC, EC, and eC prototypes.

**Data availability**. All relevant data are available from the corresponding author on reasonable request.

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

## Acknowledgements

We thank Luis Hueso, Anders Smith, Joseph Mantese, and Poppy Rowe for discussions. X.M. is grateful for support from the Royal Society. E.D., R.F., H.S., and D.S. thank the FNR in Luxembourg for funding the CO-FERMAT project through the PEARL scheme (Grant number FNR/P12/4853155/Kreisel).

## Author contributions

E.D. and N.D.M. suggested the study. E.D., G.D., and N.D.M. came up with the key idea of energy recovery displayed in Fig. 3. E.D. and S.C. ran the first set of experiments that led to Fig. 4. R.F., H.S., D.S., and E.D. ran the experiments with the prototype that led to Figs. 5 and 6. E.D., X.M., and N.D.M. interpreted the key findings. R.F., X.M., S.C., and E.D. prepared the figures. R.F. prepared the Supplementary Movie of the prototype. N.D.M. wrote the manuscript with E.D. and X.M.

## Additional information

**Competing interests:** The authors declare no competing interests.

