## [Peer Review File · Nature Communications]

Reviewers' Comments:

Reviewer #1 (Remarks to the Author)

The paper describes a method that recovers the discharged energy by transferring the electrical charges between two EC samples that work alternatively. This is an interesting work. But I do not think their findings represent a striking advance to justify publication in Nature Communications for the following two reasons.

First, the same strategy has already been exploited in pyroelectric energy harvesting. The pyroelectric effect is the converse effect of ECE. In order to enhance the energy-conversion efficiency, a large electric field synchronized with the heat exchange is applied on the pyroelectric materials (C. R. Bowen, J. Taylor, b E. LeBoulbar, D. Zabek, A. Chauhan, R. Vaish, Pyroelectric materials and devices for energy harvesting applications, Energy Environ. Sci., 2014, 7, 3836). Therefore, pyroelectric energy harvesting shares the same concerns with ECE regarding the energy recovery. As shown in a paper from R. B. Olsen (Ferroelectric Conversion of Heat to Electrical Energy-A Demonstration, J. Energy, 1982, 6, 91), a circuit that connects two pyroelectric materials together has been designed, in which the two samples alternatively work with each other and the electrical charges transfer from one to the other to recover the energy and improve the efficiency (see Figs. 4-6). The approach reported in the manuscript is essentially the same as that of Olsen's work.

Second, only a circuit for the EC materials which can transfer the charges from one material to the other is designed. Could they actually be achieved in a real device? What efficiencies are required for practical applications? What applications?

Reviewer #2 (Remarks to the Author)

In a thermodynamic closed cycle the work done in a specific leg of a cycle may be greater than the cyclic work. This is not a problem for efficiency, as the work spent on a specific branch is "recovered" in another branch. However it can be an engineering problem to appropriately "size" a power supply system. In this paper the authors argue that this sizing problem is a serious limitation to the development of electro caloric cooling. In the paper the authors propose an electronic circuit to perform the task. A journal of electronic engineering is in my opinion the appropriate context for the present study.

Response to referee 1

The paper describes a method that recovers the discharged energy by transferring the electrical charges between two EC samples that work alternatively. This is an interesting work. But I do not think their findings represent a striking advance to justify publication in Nature Communications for the following two reasons.

Thank you for writing that this is an interesting work.

First, the same strategy has already been exploited in pyroelectric energy harvesting. The pyroelectric effect is the converse effect of ECE. In order to enhance the energy-conversion efficiency, a large electric field synchronized with the heat exchange is applied on the pyroelectric materials (C. R. Bowen, J. Taylor, b E. LeBoulbar, D. Zabek, A. Chauhan, R. Vaish, Pyroelectric materials and devices for energy harvesting applications, Energy Environ. Sci., 2014, 7, 3836). Therefore, pyroelectric energy harvesting shares the same concerns with ECE regarding the energy recovery. As shown in a paper from R. B. Olsen (Ferroelectric Conversion of Heat to Electrical Energy-A Demonstration, J. Energy, 1982, 6, 91), a circuit that connects two pyroelectric materials together has been designed, in which the two samples alternatively work with each other and the electrical charges transfer from one to the other to recover the energy and improve the efficiency (see Figs. 4-6). The approach reported in the manuscript is essentially the same as that of Olsen's work.

We already explained in paragraph 2 of the main text how our work differs mechanistically from pyroelectric harvesting, but we failed to explain the conceptual difference. Recovering energy is an obvious goal when harvesting energy because the goal is to collect energy, but recovering energy is not an obvious goal when driving electrocaloric effects, partly because nobody thought of it, and partly because it was not hitherto clear whether there was much energy that could be recovered. This argument is now presented using formal language in our rewritten start of paragraph 2 in the main text, which reads:

“Electrical energy recovery has been exploited⁹ in order to improve efficiency when driving piezoelectric effects, and energy transfer has been exploited in order to improve efficiency when harvesting energy with pyroelectric materials¹⁰, but there has been no attempt to improve energy efficiency when driving EC effects, partly because this goal has not been identified, and partly because the amount of energy that may be recovered has not been calculated. Mechanistically, the energy recovery process that we demonstrate here differs from the transfer of electrical energy in piezoelectric¹¹ and pyroelectric¹⁰ harvesting, because...”

To make the story complete in this passage of text, we repeat here the issue of recovery being quantitatively worthwhile, as demonstrated later in our manuscript. You will see that we now also specify that the previous comparison with related work is mechanistic, having just explained the conceptual difference. (Also, references 10 and 11 are now swapped.)

Second, only a circuit for the EC materials which can transfer the charges from one material to the other is designed. Could they actually be achieved in a real device?

It is true that we have not built an actual cooling device, but such devices are now being realized, as summarized at the start of Ref. 13, and as will be discussed at forthcoming meetings in the UK during February 2016:

<https://royalsociety.org/events/2016/02/phase-transitions/>

http://people.ds.cam.ac.uk/xm212/campl_site/winton.shtml

Our approach will improve energy efficiency, which is of course important for any applications. Implementation in a real device is discussed in our final paragraph, with “antiphase EC effects driven by recovered energy [and] exploited in a useful refrigeration cycle [with] periodic recharging”.

What efficiencies are required for practical applications?

No value can be specified because electrocaloric coolers are likely to compete on the grounds that they can be miniaturized and switched on quickly, unlike vapour-compression systems. Therefore any improvements in efficiency will be a bonus, but there is no specific target.

What applications?

Applications for small cooling devices could include transport of medicine in the field. Applications for cooling devices that switch on quickly could include emergency cooling of electronic components that are prone to experiencing an occasional and unwanted temperature rise.

Response to referee 2

In a thermodynamic closed cycle the work done in a specific leg of a cycle may be greater than the cyclic work. This is not a problem for efficiency, as the work spent on a specific branch is "recovered" in another branch. However it can be an engineering problem to appropriately "size" a power supply system. In this paper the authors argue that this sizing problem is a serious limitation to the development of electrocaloric cooling. In the paper the authors propose an electronic circuit to perform the task. A journal of electronic engineering is in my opinion the appropriate context for the present study.

Thank you for reviewing this manuscript.

The amount by which “the work done in a specific leg of a cycle may be greater than the cyclic work” was previously not identified in the field of electrocalorics. Our manuscript shows that the extra work done can be significant. Given that this extra work is lost in every cycle, identifying recovery in terms of both need and method represents an important advance. Nature Communications has clarified that it may publish papers that report a more engineering-focused advance.

Reviewers' Comments:

Reviewer #1 (Remarks to the Author)

I remain negative regarding publication in Nature Communications as argued below. 1) I still didn't see the conceptual difference between the energy recovery devices reported in the pyroelectric devices and the EC setup shown in the manuscript. The strategies are essentially identical. As the authors pointed out, recovering energy is an obvious goal when harvesting energy. Although recovering energy has not been reported in the EC devices, it is not a surprise to adopt various techniques that are available in the pyroelectric devices (which is a much mature field) to improve energy efficiency of EC. I agree that the authors have made a good advance in the field of electrocalorics. However, I do not think their findings represent a striking advance to justify publication in Nature Communications. 2) Unfortunately and surprisingly, the authors fail to respond the rest of the comments from the Reviews 1 and 2.

Reviewer #3 (Remarks to the Author)

This paper is quite interesting and describes how to increase the efficiency of thermodynamic cooling cycles using electrocaloric materials. I think that this paper can be published after some minor revisions in Nature Communications and I will detail below the reasons taking into account the comments of previous reviewers.

1) Electrocaloric refrigeration is a "hot" topic and, since 2000 and the publication of Mischenko on giant electrocaloric properties of PZT materials, papers dealing with this topic has increased exponentially. Many electrocaloric materials are available and, for some of them, more comprehension of the physical parameters controlling the electrocaloric effect is needed. However, progress in electrocaloric refrigeration passes by driving the material in real conditions taking into account its efficiency. Results presented by Defay et al are very interesting because they detailed not only the global idea for improving the efficiency of the system but also explained how they had done it experimentally demonstrating that it is really possible.

2) The first comment of the first reviewer concerning works of Olsen. It is true that the basic idea putting two EC (or pyroelectric) elements in parallel was proposed by Olsen but the thermal and electrical times which are quite different (some order of magnitude) has not been taking into account (using inductor and diodes) as Olsen was making energy harvesting. Without the proposition of Defay et al, it is not possible to apply the idea of Olsen for cooling applications.

3) From the comment of the second reviewer, I can argue that the problem is not only from the electronic point of view but also considering the EC materials with its losses. The proposition of Defay et al would be published in an electronic journal (specialist one) if it was only an improvement of previous works but it is not the case. I confirm that this paper is highly relevant for Nature Communications.

4) I can propose some minor revisions for the papers.

First, in the introduction, maybe mechanically and electrically driving EC can be detailed for non-specialists. Here, Defay et al dealt with the switch on/off of the electric field on the EC element that can be driven electrically or mechanically as it has been developed for magnetocaloric systems. However, some papers on EC cooling systems deal with a mechanical action for thermal considerations (to put the EC material in contact with the hot and cool sinks or to make regenerative systems). For the paper of Defay et al., thermal exchanges are not considered and it can be precised.

The proposed system can control the time for charging (or discharging) the EC capacitor via the inductor and the time of thermal exchange. It can be noted that controlling the charging/discharging time of the EC element allows controlling hysteresis losses in the material and/or improving the EC activity of the material which frequency dependent. Maybe this point can be added in the last paragraph before "repeated electrocaloric effects".

Reviewer #4 (Remarks to the Author)

This paper is a demonstration of the efficiency with which the energy used in charging electrocaloric (EC) capacitors can be recovered in an isolated system. For practical EC cooling systems, the COP - the ratio of cooling power to work expended to operate the system - is expected to be an important figure of merit, in particular when EC systems are compared to thermoelectric systems. Recovering the energy used to actuate the EC effect in such a system could increase the COP.

This paper motivates the potential to improve performance based on measurements of the EC effect in isolated materials.

While this paper gives a novel framing for the experiment that cycles charge between capacitors, it is missing the larger context in terms of calculations, modeling, or other arguments to indicate how effective and significant this might be for a full EC cooling system with a cycle that transfers energy across finite temperature lifts. That broader context would make this paper much more impactful and compelling.

In this paper, the losses in the repeated cycles are attributed to Joule loss and leakage through diodes. Can the quantification of this be explained / shown more clearly with the data?

In the various assessments of cycle efficiency, it is assumed that the magnitude of the loss is the same for each cycle (as described in Supplemental Information). The inset in Figure 4(a) shows this is not a good assumption; can the non-monotonic variation shown in this inset be explained?

In the experiment, two capacitors are suspended from fine wires; how could this energy recovery be incorporated in a real device and what would be its expected efficiency?

On page 1, there is a reference ([4]--[6]) to EC prototypes that have been built; this list should also properly include the prototype described in Appl. Phys. Lett. 107, 093903 (2015) and Appl. Phys. Lett. 107, 134103 (2015).

Much of the data in Table 1 seem to be drawn directly from Ref. 13 and if so, that should be clearly cited.

The comparison in the abstract and introduction to magnetocaloric systems may not be clear to someone who is not a specialist in the field.

In its current form, this is not recommended for publication.

Response to referees

Response to Referee 1

I remain negative regarding publication in Nature Communications as argued below. 1) I still didn't see the conceptual difference between the energy recovery devices reported in the pyroelectric devices and the EC setup shown in the manuscript. The strategies are essentially identical. As the authors pointed out, recovering energy is an obvious goal when harvesting energy. Although recovering energy has not been reported in the EC devices, it is not a surprise to adopt various techniques that are available in the pyroelectric devices (which is a much mature field) to improve energy efficiency of EC. I agree that the authors have made a good advance in the field of electrocalorics. However, I do not think their findings represent a striking advance to justify publication in Nature Communications.

We have three points of novelty with respect to previous work in pyroelectrics:

- 1) We reveal via Table 1 that a lot of the work done to drive EC effects in some useful cycle is available for recovery. This was not known, and if there were very little work to recover then its recovery would be less exciting. (Our previous version failed to remind the reader of this point in the concluding paragraph, but that is now rectified.)
- 2) Retrospectively, it is indeed not a surprise to adopt the techniques used for pyroelectric devices to improve energy efficiency of EC devices, but nobody has previously suggested this idea.
- 3) As noted in point 2) of Referee 3, our work differs from recovery in pyroelectrics because we have reconciled the different thermal and electrical time scales that are relevant for EC devices. (Our previous version failed to remind the reader of this point in the concluding paragraph, or paragraph two where we discuss pyroelectric harvesting, but that is now rectified.)

Note that the importance of energy recovery has been recognised by:

- Qiming Zhang in his February 2016 talk on electrocalorics at the Royal Society – he may well have learned about energy recovery from an earlier talk given in Slovenia by our first author Emmanuel Defay.
- Andrej Kitanovski, who quoted in his tutorial at MRS Spring 2016 the 80% recovery presented previously by Emmanuel Defay.

2) Unfortunately and surprisingly, the authors fail to respond the rest of the comments from the Reviews 1 and 2.

We submitted a point-by-point response to all points by both referees, so we do not understand this comment. Is it possible that the referee was not sent any of our responses? We are asking the Editor to check on this.

Overall: the paper has been rewritten in many places to improve presentation, to highlight the novelty in the concluding paragraph, and to make the improvements suggested by all referees.

Response to Referee 3

This paper is quite interesting and describes how to increase the efficiency of thermodynamic cooling cycles using electrocaloric materials. I think that this paper can be published after some minor revisions in Nature Communications and I will detail below the reasons taking into account the comments of previous reviewers.

1) Electrocaloric refrigeration is a "hot" topic and, since 2000 and the publication of Mischenko on giant electrocaloric properties of PZT materials, papers dealing with this topic has increased exponentially. Many electrocaloric materials are available and, for some of them, more comprehension of the physical parameters controlling the electrocaloric effect is needed. However, progress in electrocaloric refrigeration passes by driving the material in real conditions taking into account its efficiency. Results presented by Defay et al are very interesting because they detailed not only the global idea for improving the efficiency of the system but also explained how they had done it experimentally demonstrating that it is really possible.

2) The first comment of the first reviewer concerning works of Olsen. It is true that the basic idea putting two EC (or pyroelectric) elements in parallel was proposed by Olsen but the thermal and electrical times which are quite different (some order of magnitude) has not been taking into account (using inductor and diodes) as Olsen was making energy harvesting. Without the proposition of Defay et al, it is not possible to apply the idea of Olsen for cooling applications.

3) From the comment of the second reviewer, I can argue that the problem is not only from the electronic point of view but also considering the EC materials with its losses. The proposition of Defay et al would be published in an electronic journal (specialist one) if it was only an improvement of previous works but it is not the case. I confirm that this paper is highly relevant for Nature Communications.

Thank you for the above comments, especially your much-appreciated defence of our work in respect of the first two referees.

4) I can propose some minor revisions for the papers.

First, in the introduction, maybe mechanically and electrically driving EC can be detailed for non-specialists. Here, Defay et al dealt with the switch on/off of the electric field on the EC element that can be driven electrically or mechanically as it has been developed for magnetocaloric systems. However, some papers on EC cooling systems deal with a mechanical action for thermal considerations (to put the EC material in contact with the hot and cool sinks or to make regenerative systems). For the paper of Defay et al., thermal exchanges are not considered and it can be precised.

Excellent idea. We now have a paragraph in the introduction that reads as follows:

“EC effects are typically parameterized in terms of adiabatic temperature change ΔT , or isothermal entropy change ΔS . Cooling devices may exploit EC effects near either of these two thermal limits, or somewhere in between. The heat associated with driving an EC capacitor is ultimately dumped to a sink, while the heat associated with undriving the same capacitor is ultimately absorbed from a load, resulting in a net transfer of heat from load to sink. EC capacitors, which are thus driven and undriven, may be alternately connected to sinks and loads by mechanical means⁴⁻⁷, or moved to each end of a regenerator¹⁶ that thus develops a large temperature span along its length⁸⁻⁹.”

The proposed system can control the time for charging (or discharging) the EC capacitor via the inductor and the time of thermal exchange. It can be noted that controlling the charging/discharging time of the EC element allows controlling hysteresis losses in the material and/or improving the EC activity of the material which frequency dependent. Maybe this point can be added in the last paragraph before "repeated electrocaloric effects".

Excellent idea, but as it is for the future, we have worked it into our concluding paragraph rather than the relevant paragraph on results. We write:

“In future, one could tune the speed of charge transfer to reduce losses in frequency-dependent relaxor ferroelectrics^{7,21-23,}”

Overall: the paper has been rewritten in many places to improve presentation, to highlight the novelty in the concluding paragraph, and to make the improvements suggested by all referees.

Response to Referee 4

This paper is a demonstration of the efficiency with which the energy used in charging electrocaloric (EC) capacitors can be recovered in an isolated system. For practical EC cooling systems, the COP - the ratio of cooling power to work expended to operate the system - is expected to be an important figure of merit, in particular when EC systems are compared to thermoelectric systems. Recovering the energy used to actuate the EC effect in such a system could increase the COP.

This paper motivates the potential to improve performance based on measurements of the EC effect in isolated materials.

Thank you for these positive comments.

While this paper gives a novel framing for the experiment that cycles charge between capacitors, it is missing the larger context in terms of calculations, modeling, or other arguments to indicate how effective and significant this might be for a full EC cooling system with a cycle that transfers energy across finite temperature lifts. That broader context would make this paper much more impactful and compelling.

Excellent idea. We have introduced Table 2, described in a new penultimate paragraph, in order to show how COP values for the six EC prototypes would be improved with 80% energy recovery. The referee will know that COP is maximized for zero temperature lift ($T_h = T_c$), and zero for maximum temperature lift. We have therefore worked with maximum COP values. These were not specified in each report, and had to be calculated. Data are not available for COPs at finite temperature lifts. More generally, improvements to COPs are expected to be large given that there should be plenty of recoverable work that is not consumed in the thermodynamic cycle (Table 1).

In this paper, the losses in the repeated cycles are attributed to Joule loss and leakage through diodes. Can the quantification of this be explained / shown more clearly with the data?

Joule heating during our 30 s wait times is quantified and explained by changing:
“an exceptionally low leakage current (<1 nA at 200 V) implies negligible Joule heating while our charged MLCs shed EC heat”

to

“an exceptionally low leakage current implies negligible Joule heating while our charged MLCs shed EC heat (<1 nA at 200 V over 30 s implies <5 mK), consistent with the return to near ambient (see Fig. 4b, later)”

We have quantified the small leakage through diodes by changing:

“the convergence of the plateaux was also due in part to a small degree of leakage through the active diode while waiting for heat flow between transfers, as evidenced by the small rise in the low-voltage plateaux that accompanies the small fall in the high-voltage plateaux (this redistribution of charge is barely perceptible in Fig. 4a).”

to

“the convergence of the plateaux was also due in part to a small degree of leakage through the active diode while waiting for heat flow between transfers, as evidenced by the small rise

in the low-voltage plateaux that accompanies the small fall in the high-voltage plateaux (this redistribution of charge is barely perceptible in Fig. 4a; it is largest at the outset where between $t = 0$ and 30 s it reduces the high-voltage plateau by a maximum of 2.1%).”

In the various assessments of cycle efficiency, it is assumed that the magnitude of the loss is the same for each cycle (as described in Supplemental Information). The inset in Figure 4(a) shows this is not a good assumption; can the non-monotonic variation shown in this inset be explained?

Good question. We have inserted the following paragraph to explain this:

“The losses that limit the initial values of $\eta_i^\Delta/\eta_i \sim 4$ and $\alpha_i \sim 0.75$ are eventually reduced to a value (Supplementary Note 4) that is compatible with the 3 A diode rating, such that these two parameters are maximised for transfer $i = 6$, both here (inset, Fig. 4a) and in every trial we performed. The transfer losses are diminished in subsequent transfers because the MLCs exchange less energy, but there is nevertheless a fall in α_i and η_i^Δ/η_i , because of proximity to the 0.7 V diode threshold, and because of leakage losses during the ~ 30 s wait times.”

Supplementary Note 4 is as follows:

In the experiment, two capacitors are suspended from fine wires; how could this energy recovery be incorporated in a real device and what would be its expected efficiency?

Our circuit applies to MLCs operating in antiphase. We did not previously explain the thermal exchange required for a real device, but this is now explained in a new paragraph 3 that reads:

“EC effects are typically parameterized in terms of adiabatic temperature change ΔT , or isothermal entropy change ΔS . Cooling devices may exploit EC effects near either of these two thermal limits, or somewhere in between. The heat associated with driving an EC capacitor is ultimately dumped to a sink, while the heat associated with undriving the same capacitor is ultimately absorbed from a load, resulting in a net transfer of heat from load to sink. EC capacitors, which are thus driven and undriven, may be alternately connected to sinks and loads by mechanical means⁴⁻⁷, or moved to each end of a regenerator¹⁶ that thus develops a large temperature span along its length⁸⁻⁹.”

The expected efficiency is treated in terms of COP values described earlier in this response.

On page 1, there is a reference ([4]--[6]) to EC prototypes that have been built; this list should also properly include the prototype described in Appl. Phys. Lett. 107, 093903 (2015) and Appl. Phys. Lett. 107, 134103 (2015).

Yes, Appl. Phys. Lett. 107, 134103 (2015) is now included as ref. 9, and in fact we now cite different EC prototypes in refs 4-9. But the other paper [Appl. Phys. Lett. 107, 093903] describes calculations in respect of [Appl. Phys. Lett. 107, 134103 (2015)] rather than the experiment itself, so we should not cite this when listing EC prototypes.

Much of the data in Table 1 seem to be drawn directly from Ref. 13 and if so, that should be clearly cited.

Ref. 13 is now ref. 18 in the revised manuscript. We now cite it in the table caption when referring to the evaluation of the work. Note that other values drawn from elsewhere come directly from the original reference cited in the table.

The comparison in the abstract and introduction to magnetocaloric systems may not be clear to someone who is not a specialist in the field.

We have rewritten this and many other parts of the paper.

In its current form, this is not recommended for publication.

Overall: the paper has been rewritten in many places to improve presentation, to highlight the novelty in the concluding paragraph, and to make the improvements suggested by all referees.

Reviewers' Comments:

Reviewer #4 (Remarks to the Author)

This paper is written more clearly and comprehensively in its current version. It is clear that energy recovery could be important for EC systems, operation of EC components in antiphase could be an important enabler, and that it is critical to consider thermal and electrical time scales to make a working device.

It is notable that the framing of this paper is a comparison to magnetocaloric systems, which exist as functional prototypes. Indeed, this work is important not at a fundamental physics or materials level but rather in how it can enable practical systems. This framing begs the question of how energy recovery could be implemented in a full system and the impact that it would have beyond the first test of an isolated MLC.

As one example from the paper: the authors note that while an Ericsson cycle is desirable, practical considerations demand instead Brayton cycle operation. Other practical considerations in building a real system that operates at finite temperature lift ($T_h > T_c$) could significantly change the possibility of and impact of energy recovery. Just as the temperature lift of an isolated film of EC material is greater than the temperature lift achievable in an MLC, the COP of a complete system is expected to be different from that of an isolated MLC. For example, it may be that the COP is reduced so much by other effects (such as the energy needed for actuation) that the energy recovery makes only a small difference in a full system.

It is recognized that extrapolation to a real system is limited in part by the lack of reported values of COP in the literature. Table 2 shows a creative way to summarize existing literature and may be misleading. There is no argument made in this paper as to why the 5x improvement in COP from an isolated MLC is extensible to a full system or to a system that operates at finite temperature lift.

This paper shows a demonstration of a first step in understanding energy recovery in electrocaloric materials. Some reasoning and/or additional experiments to illustrate the robustness and extensibility of the results and how energy recovery might work in a heat pump with finite temperature lift would help complete the comparison to magnetocalorics, and show the reader how this work can advance the very applied field of electrocaloric cooling systems.

Response to Reviewer #4

We thank the reviewer for the positive comments, and for inspiring us to demonstrate energy recovery in a prototype that we now describe in the manuscript.

This paper is written more clearly and comprehensively in its current version. It is clear that energy recovery could be important for EC systems, operation of EC components in antiphase could be an important enabler, and that it is critical to consider thermal and electrical time scales to make a working device.

Thank you for agreeing that energy recovery is important. Apart from adding the information about the prototype, the rest of the paper has been rewritten completely to present everything much better.

It is notable that the framing of this paper is a comparison to magnetocaloric systems, which exist as functional prototypes. Indeed, this work is important not at a fundamental physics or materials level but rather in how it can enable practical systems. This framing begs the question of how energy recovery could be implemented in a full system and the impact that it would have beyond the first test of an isolated MLC.

This is the comment that inspired the work with the prototype.

As one example from the paper: the authors note that while an Ericsson cycle is desirable, practical considerations demand instead Brayton cycle operation. Other practical considerations in building a real system that operates at finite temperature lift ($T_h > T_c$) could significantly change the possibility of and impact of energy recovery. Just as the temperature lift of an isolated film of EC material is greater than the temperature lift achievable in an MLC, the COP of a complete system is expected to be different from that of an isolated MLC. For example, it may be that the COP is reduced so much by other effects (such as the energy needed for actuation) that the energy recovery makes only a small difference in a full system.

We present a COP for the prototype we now demonstrate, but we are careful to explain that this COP “is not a true device COP for two reasons. First, heat Q is only pumped from our load towards our sink in order to compensate for heat leaking into our load from the environment, but with appropriate thermal isolation one could instead pump heat Q from a target item rather than the environment. Second, work W does not include the work done to drive the motor, the power supplies and the micro-controller, because we have not minimised the avoidable component of this work by designing bespoke components (this type of omission is common practice for MC, EC and eC prototypes).”.

We also qualify our use of COP elsewhere in the paper:

1. by writing in our abstract that “energy recovery reduces the net work done on the electrocaloric material by a factor of 3.4, increasing the corresponding coefficient of performance from 2.9 to 10.1”.

2. by writing at the end of the introduction “Last, we demonstrate a prototype EC refrigerator in which energy recovery enhances our as-defined coefficient of performance (COP) by a factor of 3.4.”.
3. by writing in our summary paragraph that we have “energy recovery in a prototype heat pump for which we define a COP that is increased by a factor of up to 3.4”.

We are grateful to the referee for getting us to explain properly what we mean by COP.

It is recognized that extrapolation to a real system is limited in part by the lack of reported values of COP in the literature. Table 2 shows a creative way to summarize existing literature and may be misleading. There is no argument made in this paper as to why the 5x improvement in COP from an isolated MLC is extensible to a full system or to a system that operates at finite temperature lift.

Assuming that the factor of 5 would apply to prototypes was speculation. Now that we have the prototype, this speculation is not helpful, and Table 2 is deleted.

This paper shows a demonstration of a first step in understanding energy recovery in electrocaloric materials. Some reasoning and/or additional experiments to illustrate the robustness and extensibility of the results and how energy recovery might work in a heat pump with finite temperature lift would help complete the comparison to magnetocalorics, and show the reader how this work can advance the very applied field of electrocaloric cooling systems.

We have now demonstrated energy recovery with a prototype, as requested by the referee. Once again, we thank the referee for inspiring us to do this.

Reviewers' Comments:

Reviewer #4:

Remarks to the Author:

It is noted that in this version, there is additional demonstration via a lab-scale prototype. This allows a discussion and demonstration of the tradeoffs between things like temperature lift and the effectiveness of energy recovery, as shown in Fig. 6. With this addition, the paper is much stronger.

In general, the notation and accounting of efficiency in this paper is challenging to follow. In this last section, there is a critical clarification that is missing. It is not obvious if the various figures of merit shown in Fig. 6 are deduced from when the system is in steady state and it is not clear if they are drawn from measurements averaged over a number of cycles or just picked from one half cycle. As the earlier work shown in Fig. 4 demonstrates, any figure of merit picked from one cycle may not be an appropriate description of the steady-state performance of the system.

For clearly tying the analysis of the prototype and Fig. 6 to the earlier work shown in Fig. 4, it is necessary to derive the figures of merit like WR from averages over many cycles after the system has reached steady state (ie past the point shown in Fig. 5). Without this, it is not clear that the results are appropriately represented.

Other notes for clarity:

(1) There is a parenthetical note of why $WR(V^{O_{12C2}}=0) = 63\%$ and not 71%. Given the non-standard accounting of energy here, this could be explained a more clearly.

(2) The video has the potential to be a nice illustration. It is hard to map the video onto the schematic in Fig. 5. To realize the video's full impact, I suggest text or another schematic that explains what the different components are in the video and a sentence that summarizes what the viewer is meant to take away from the video.

As is, this is not yet recommended for publication. If the analysis of the prototype can be presented with measurements that are clearly drawn from averages in the steady-state, then this work can be a more clear contribution to the field.

Response to Reviewer #4

We thank the Referee very much indeed for comments that have inspired us to greatly improve the presentation of our work. The main change is that we have completely rewritten our description of the prototype in the main text and Methods. We have also deleted Fig. 6 (as explained below) and made some other minor changes. Fig. 5 is improved, and all other changes are highlighted in yellow. We apologise for the poor presentation of our original work, and would be grateful if the Referee can please consider the revised files.

It is noted that in this version, there is additional demonstration via a lab-scale prototype. This allows a discussion and demonstration of the tradeoffs between things like temperature lift and the effectiveness of energy recovery, as shown in Fig. 6. With this addition, the paper is much stronger.

The construction of this prototype was inspired by the Referee's comments in the previous round, and we are grateful for them.

In general, the notation and accounting of efficiency in this paper is challenging to follow. In this last section, there is a critical clarification that is missing. It is not obvious if the various figures of merit shown in Fig. 6 are deduced from when the system is in steady state and it is not clear if they are drawn from measurements averaged over a number of cycles or just picked from one half cycle. As the earlier work shown in Fig. 4 demonstrates, any figure of merit picked from one cycle may not be an appropriate description of the steady-state performance of the system.

For clearly tying the analysis of the prototype and Fig. 6 to the earlier work shown in Fig. 4, it is necessary to derive the figures of merit like WR from averages over many cycles after the system has reached steady state (ie past the point shown in Fig. 5). Without this, it is not clear that the results are appropriately represented.

Presentation

We very much agree with the Referee that our previous presentation was challenging to follow. We have performed a complete rewrite because there were so many problems with our previous effort, e.g. (1) we explained the basic principle of operation by describing both MLC plates in one half cycle (with details of timing), rather than one MLC plate in a complete cycle (without details of timing), and (2) we only explained the energy recovery circuit for the prototype in Methods (with details of components), rather than in the main text (with details of components in Methods).

Steady-state operation

Our failure to explain the issue of steady-state operation was yet another problem. We now explain that the adiabatic electrocaloric cooling in our MLC plates is similar at any time after turn on (sharp drops in green and purple data, revised Fig. 5b). To clarify that the parameters we present describe steady-state operation arising ~250 s after turn on, we have added this clarification at seven points in the main text, and extended the time interval that we show in Fig. 5b.

Fig. 6

As we now say in the manuscript, our main result is that we use energy recovery to improve the COP of our device operating in the steady state (Fig. 5) by a factor of 3.4. Fig. 6 was

included to demonstrate how we might optimize this performance, and it described how the heat and work associated with the adiabatic electrocaloric cooling in one of our MLC plates would be affected by setting a finite voltage rather than zero voltage on the other plate. Given that these adiabatic electrocaloric effects are the same both before and after reaching the steady state, these one-off experiments represent what would be achieved in steady-state operation. However, we could not in practice achieve continuous operation with the finite voltage, and the complicated addition only led us to infer a very small increase of COP (from 10.1 to 11.7), so Fig. 6 and its description are deleted.

Other notes for clarity:

(1) There is a parenthetical note of why $WR(V_{12C2}=0) = 63\%$ and not 71%. Given the non-standard accounting of energy here, this could be explained a more clearly.

The deletion of Fig. 6 has led to the deletion of the text containing this note.

(2) The video has the potential to be a nice illustration. It is hard to map the video onto the schematic in Fig. 5. To realize the video's full impact, I suggest text or another schematic that explains what the different components are in the video and a sentence that summarizes what the viewer is meant to take away from the video.

The improved video now has colour overlays that are consistent with the Fig. 5a schematic, and we have the following caption to describe what is shown:

Description of Movie 1

The IR camera data show that plates 12C1 and 12C2 take it in turns to make contact with the load, undergo adiabatic electrocaloric cooling, and absorb heat from both the load and the voltage leads. Away from the field of view, each plate makes contact with its sink and dumps heat. The load is eventually seen to reach a steady-state temperature that is limited by heat leaks. Colour overlays denote the regions whose average temperature we report.

As is, this is not yet recommended for publication. If the analysis of the prototype can be presented with measurements that are clearly drawn from averages in the steady-state, then this work can be a more clear contribution to the field.

We have now clarified how data of interest describe steady-state operation. To summarize the novelty of our work:

1. We have demonstrated that there can be a large amount of energy to recover when driving and undriving electrocaloric effects (Table 1).
2. We have demonstrated how to recover this energy and reconcile the very different thermal and electrical timescales (Figs 3,4).
3. We have experimentally demonstrated how to implement energy recovery in an electrocaloric prototype cooling device that operates in the steady state (Fig. 5), such that the coefficient of performance is increased by a factor of 3.4 from 2.9 to 10.1.

Reviewers' Comments:

Reviewer #4:

Remarks to the Author:

This work is of interest to the community.

A note on the results: It seems strange to define COP as (average Q / average W) instead of defining an average (Q/W) .

The references now deserve an update in this paper.

Reviewer #4:

This work is of interest to the community.

We thank the referee for this positive comment.

A note on the results: It seems strange to define COP as (average Q / average W) instead of defining an average (Q/W).

We now explain this issue implicitly in the main text where we define our COP, and explicitly in Methods.

In the main text, our revised COP definition reads as follows:

“For many cycles of steady-state operation, where heat Q was pumped from the load in each half cycle by doing work W in the previous half cycle, we will identify the COP for our prototype as the total heat pumped from the load divided by the total work done, such that $\text{COP} = \overline{Q}/\overline{W}$. (This is discussed further in Methods, along with the fact that COPs calculated from values of heat and work represent upper bounds on device COPs.)”

In Methods, our explicit answer to the question reads as follows:

“Given that work W is done in one half cycle (EC heating) to pump heat Q in the next half cycle (EC cooling), there would be an ambiguity if one were to calculate $\text{COP} = \overline{Q}/\overline{W}$. This is because one could choose to divide values of Q and W that describe either the same half cycle and different plates, or different half cycles and the same plate.”

The references now deserve an update in this paper.

We have added these two references on new electrocaloric prototypes:

- [12] Zhang, T., Qian, X.-S., Gu, H., Hou, Y., & Zhang, Q.M. An electrocaloric refrigerator with direct solid to solid regeneration. *Appl. Phys. Lett.* **110**, 243503 (2017).
- [13] Ma, R., Zhang, Z., Tong, K., Huber, D., Kornbluh, R., Sungtaek Ju, Y. & Pei, Q. Highly efficient electrocaloric cooling with electrostatic actuation. *Science* **357**, 1130–1134 (2017).

... and these two references on new mechanocaloric prototypes:

- [17] Tušek, J., Engelbrecht, K., Eriksen, D., Dall'Olio, S., Tušek J. & Pryds, N. A regenerative elastocaloric heat pump. *Nature Energy* **1**, 16134 (2016).
- [18] Ossmer, H., Wendler, F., Gueltig, M., Lambrecht, F., Miyazaki S. & Kohl, M. Energy-efficient miniature-scale heat pumping based on shape memory alloys. *Smart Mater. Struct.* **25**, 085037 (2016).